# Hope Mediates Stress to Reduce Burden in Family Caregivers of Persons with Alzheimer’s Disease

**DOI:** 10.3390/geriatrics9020038

**Published:** 2024-03-18

**Authors:** Jocelyn Shealy McGee, Edward C. Polson, Dennis R. Myers, Angela McClellan, Rebecca Meraz, Weiming Ke, Holly Carlson Zhao

**Affiliations:** 1Garland School of Social Work, Baylor University, Waco, TX 76701, USA; clay_polson@baylor.edu (E.C.P.); dennis_myers@baylor.edu (D.R.M.); ammcclel@central.uh.edu (A.M.); 2Louise Herrington School of Nursing, Baylor University, Dallas, TX 75246, USA; rebecca_meraz@baylor.edu (R.M.); weiming_ke@baylor.edu (W.K.); 3Center for Optimal Brain Health, Houston, TX 77057, USA; hcarlson@cfobh.com

**Keywords:** dementia, caregiving, stress, burden, hope, positive psychology

## Abstract

The experience of burden among family caregivers of persons with Alzheimer’s disease and other forms of dementia may be deleterious for their health and well-being. Little is known, however, about the degree to which internal positive psychological resources, such as hope, influence burden perceptions in this population. The current study is novel in that it examined how multiple dimensions of hope, hope–agency and hope–pathway, influenced burden in a sample of one-hundred and fifty-five family caregivers of persons with Alzheimer’s disease. The stress process model was used as the theoretical framework for variable specification in this study. Hope was conceptualized using Snyder and colleagues’ hope theory. Supporting our first hypothesis, we found that burden was negatively associated with hope–agency, r = −0.33, *p* < 0.001 and hope–pathway, r = −0.24, *p* < 0.01. Multiple regression was used to determine if hope–agency and hope–pathway independently contributed to burden. Analysis revealed that hope–agency but not hope–pathway influenced burden when other key variables were taken into consideration. Findings from mediation analysis affirmed that hope–agency had a small but significant mediation effect between stress and burden in this sample. This study provides evidence for the relevance of assessing multiple dimensions of hope when working with caregivers of persons with Alzheimer’s. Although replication studies are warranted, the current study confirms a need for further development and refinement of hope-bolstering behavioral interventions which may mediate stress and burden in this population. These interventions should be systematically assessed for efficacy and effectiveness via implementation studies in real-world settings.

## 1. Caregiver Stress and Burden

An estimated 55.2 million people around the world have dementia which is expected to increase to 78 million by the year 2030 [1], with Alzheimer’s disease being the most prevalent form of progressive age-related dementia [2]. Prior research has focused mainly on the stressors associated with caregiving which can lead to a sense of burden and poor outcomes for health and well-being in family caregivers [3]. Less is known, however, about the influence of internal positive psychological resources (i.e., character strengths and virtues) [4], such as hope, which may possibly serve to reduce or mitigate burden in this population leading to better outcomes. Therefore, the current study examined the influence of hope, as a multidimensional resource, for reducing burden in a sample of family caregivers of persons Alzheimer’s. An overview of the published research on caregiver stress and burden, hope theory, and hope and caregiving are provided as background information for the study.

Burden among family caregivers has been associated with physical (e.g., sleep difficulties, cardiovascular disease, reduced immune response, etc.), emotional (e.g., mental health difficulties), social and lifestyle changes (e.g., relationship challenges, lower social engagement) [5,6], and spiritual struggles [7,8]. Based on meta-analyses, prevalence rates of thirty-one percent for depression [9] and forty-two percent for anxiety [10] have been predicted among caregivers of persons with dementia. Contributing to emotional health concerns, caregivers may limit their own self-care activities, given the demands on their time, which may lead to interrupted life goals and appraising the role of caregiver as burdensome [11].

Additionally, it has been suggested that caregiver burden may impact mortality rates in persons with dementia [12], although the reasons are not clear. Preliminary evidence suggests that the caregivers of persons with dementia who do not receive community-based social support (i.e., homemaker and home health services) may experience greater burden, making time to placement in long-term care settings such as nursing homes sooner than the caregivers who receive this support [13]. Some research suggests that there are higher mortality rates among persons with dementia the first year they are placed in a long-term care setting as compared to living in their own home [14,15].

There are several sociodemographic factors that may influence burden perceptions in caregivers such as age, gender, and the relationship between the caregiver and the care recipient. More specifically, caregivers in mid-life and between the ages of 65 and 74 tend to have more responsibilities outside of caregiving than those who are over the age of 75, which may lead to a greater sense of burden [16]. Additionally, caregiver gender may influence burden perceptions, with those identifying as female reporting more burden than those identifying as male [17,18]. Research also suggests that wives tend to experience greater burden than husbands [19]. Perceived social support is a protective factor against burden, based on a recent meta-analysis [20] as may be spirituality and faith [7,8,21].

## 2. Hope Theory

Snyder’s [22] hope theory is perhaps the most robust and well-researched model of hope. This theory conceptualizes hope as an active process comprised of two dimensions which work in synergy to empower and encourage persons amid stressful life circumstances [22]. One dimension of hope, hope–agency (“I can do this.”) refers to the extent to which a person believes they have the capacity to progress towards an intended goal [23]. A second dimension of hope, hope–pathway (“How can I do this?”) is the extent to which a person believes they know the means or methods for attaining their goal [23]. The efficacy of independent hope–agency and hope–pathway dimensions have been validated [24]. Hope–agency, which has to do with self-efficacy and a sense of personal control, may be a more robust predictor than hope–pathway for coping [25] and has demonstrated the capacity to uniquely predict positive coping in specific stressful situations [26].

## 3. Hope and Caregiving

There is evidence that hope is a protective resource against burden in caregivers of people with dementia when compared to other positive psychological factors such as social intelligence, zest, and love [27]. Additionally, greater levels of hope in caregivers are associated with fewer symptoms of depression [28]. The theory of “Renewing Everyday Hope” [29], which emerged from a grounded theory qualitative study, posits that hope among caregivers can be likened to an interconnected ‘knot’ that involves coming to terms with a loved one’s condition as well as the caregivers’ own lived experiences, finding positives in these circumstances by weighing the pros and cons, connecting with others and faith, and seeing possibilities for the future by setting goals and making choices.

## 4. The Current Study

The purpose of the current study was to examine the relationship between stress, the two dimensions of hope (hope-agency and hope-pathway), and perceived burden in a sample of family caregivers of persons with dementia. The caregiving stress process model [30,31] provided the framework for variable selection and specification aimed at addressing the complexity of the caregiving experience. Stress, in this model, is categorized as primary or secondary, with primary stress involving the objective demands of caregiving (i.e., the amount of physical care required, the degree of cognitive impairment in the care recipient, etc.). Secondary stress accounts for sources of strain not directly related, but consequential, to the caregiving role such as contextual or background variables such as the sociodemographic profiles of caregivers and care recipients. Internal positive psychosocial resources, such as hope, are considered, and may directly or indirectly contribute, to reducing the untoward effects of caregiver stress, which can lead to burden. External resources (i.e., social support) may also mediate or moderate relationships among contextual factors, stressors, and caregiver outcomes in this model.

The aims of the current study were to: (1) examine the degree to which hope–agency and hope–pathway are related to perceived burden among family caregivers of persons with Alzheimer’s disease; (2) examine if hope–agency and hope–pathway independently and significantly contribute to explaining variation in burden in this sample; and (3) determine if hope–agency and hope–pathway independently and significantly mediate the relationship between stress and burden in this sample.

## 5. Method

### 5.1. Study Design

A cross-sectional quantitative research design was used in this study with a convenience sample in a large metropolitan area in the Southwestern part of the United States. This research project was approved by the Baylor College of Medicine and Baylor University Institutional Review Boards prior to any study activities. The study was open to adults 21 years and older who self-identified as a family caregiver of a person diagnosed with probable Alzheimer’s disease who was aged 65 or older and in the early stages of disease progression. The ability to read and speak English was an inclusion criterion for participants. Caregivers who had a diagnosis of any form of dementia or a serious untreated mental illness were excluded from the study.

Recruitment strategies involved the distribution of flyers about the study to organizations who serve people with Alzheimer’s disease and other forms of dementia and their family members, as well as community presentations about the study to professionals (i.e., chaplains, nurses, physicians, psychologists, social workers, etc.) who serve this population. After screening for eligibility, potential participants went through an informed consent process. Upon consent, they were invited to complete a packet of self-report measures in the English language.

Sample size was based on a priori power analysis to find significance with a desired power of 0.80, an α-level of 0.05, and a moderate–small effect size of 0.15 (f2). For a multiple linear regression model with seven predictor variables, the minimum sample size needed was 103. Thus, the obtained sample size of 155 participants was adequate.

### 5.2. Variables and Measures

Standardized measures were used to assess the variables of hope-agency, hope-pathway, caregiver burden, and stress while intensity of care, social support, and demographic/background were measured with investigator developed measures. Variables and corresponding measures are described below.

#### 5.2.1. Outcome Variable

Caregiver Burden. Caregiver burden was measured using the Zarit Burden Inventory [32]. This 22-item, self-report tool assesses the degree of perceived burden among adult caregivers. Response options are presented on a 5-point, Likert Scale, ranging from 0 (Never) to 4 (Nearly Always). This measure demonstrated good to excellent internal consistency in prior studies [33]. Excellent internal consistency was demonstrated in the current study (Cronbach α = 0.93).

#### 5.2.2. Predictor Variables

Hope. The 12-item Adult Hope Scale [34] was used to measure caregivers’ internal resources using two dimensions of hope: hope–agency and hope–pathway (which were discussed in the background section of this article). This measure has demonstrated adequate reliability for each dimension in previous studies [34]. In the current study, good internal consistency for the hope–agency subscale, Cronbach’s α = 0.82; and the hope–pathway subscale, Cronbach’s α = 0.84 was demonstrated.

Caregiver Stress. The distress subscale of the Neuropsychiatric Inventory Questionnaire [35] was used to assess participants’ subjective stress. The distress subscale measures the degree to which caregivers are distressed by the presence, severity, and frequency of 12 challenging behaviors associated with dementia. Content validity, concurrent validity, inter–rater reliability, and test–retest reliability of the NPI distress subscale is well established [35]. This subscale demonstrated good internal consistency in this study (Cronbach’s α = 0.85).

Intensity of Care. The investigators developed an Intensity of Care index to objectively assess participants’ objective stress. This additive index was created by summing responses to two survey items: (1) the average number of hours per day spent providing direct care for the person with dementia; and (2) the number of people cared for (which could also include other family members such as children).

Secondary Stressors. The measures used for secondary stressors were caregiver age, gender, and marital status. These data were gathered using a series of demographic self-report items.

Social Support. The external resource of social support was assessed through the use of a clinical measure that was part of an intake packet at the center where the study was conducted. This measure was comprised of 4 items using a 5-point, Likert Scale, ranging from Strongly Agree to Strongly Disagree. The higher the score on this measure, the greater the social support. Although this social support measure is not standardized, it demonstrated adequate internal consistency in the current study (Cronbach α = 0.78).

### 5.3. Data Analysis

Data were analyzed using SPSS version 27. To examine the aims of the study, we first calculated descriptive statistics for participants’ demographics/background variables and the other variables of interest. Next, we analyzed correlations between the outcome variable (burden), the measures for stress, objective and subjective primary stressors, secondary stressors, and measures for external and internal resources. Third, we developed a series of stepwise hierarchical regression models to examine the independent effect that each dimension of hope had on explaining variation in caregiver burden when controlling for primary objective and subjective stressors, secondary stressors, and social support as an external resource. Additional diagnostic analyses assessed for multicollinearity between the independent variables in the regression models. Finally, mediation models were constructed to test whether hope–agency had a mediating effect on the relationship between our measure of subjective stress and burden. Since the Sobel test is conservative [36], an increasingly popular method of testing the indirect effect, bootstrapping, was used [37].

## 6. Results

### 6.1. Sample Demographics and Descriptive Statistics

One hundred and fifty-five caregivers were included in the study. The average age of participants was 65 years (*SD* = 11.06). Most identified as female (70.6%; *n* = 108) and as White or Caucasian (94.8%, *n* = 145). A majority indicated that they were married (90.8%, *n* = 138), and the largest group of participants reported being spousal caregivers (66.2%, *n =* 100). A smaller group reported being an adult child or grandchild (23.2%, *n =* 35). Table 1 provides additional demographic data, and Table 2 provides a descriptive profile of the sample’s responses for key variables that were under analysis in the current study.

### 6.2. Correlates of Burden

To address the first aim of the study, correlates of burden were examined. Perceived burden was positively related to objective stress, r = 0.23, *p* < 0.01; subjective stress, r = 0.62, *p* < 0.001; and caregiver gender, r = 0.27, *p* < 0.01. There was a significant negative association between burden and the external resource of social support, r = −0.22, *p* < 0.05. There were no correlations between burden and age or marital status. Supporting our hypothesis, we found that burden was negatively associated with the hope–agency, r = −0.33, *p* < 0.001 and hope–pathway scales, r = −0.24, *p* < 0.01. See Table 3.

### 6.3. Independent Effects of Hope Measures on Caregiver Burden

To address the second aim of the study, a series of stepwise hierarchical regression models for predicting burden were developed. We tested for a significant change in *R*^2^ (∆*R*^2^) between the base model and models for each of the hope dimensions.

Model 1 served as the base model and included measures of subjective stress, primary and secondary objective stress, and the external resource of social support. The base model revealed that subjective stress (*β* = 0.566, *p* < 0.001) and caregiver gender (*β* = 0.266, *p* < 0.001) were significantly related to burden. In other words, caregivers who identified as female with higher levels of subjective stress also had higher levels of burden. Standardized betas revealed that subjective stress was the strongest predictor in the base model for burden.

For models 2 and 3, measures of hope–agency and hope–pathway were introduced in a stepwise fashion to test whether each of these dimensions had an independent effect on burden. Results for model 2 suggested that the inclusion of hope–agency improved the fit of the model and accounted for a 4.93% increase in the explained variance in burden. Further, hope–agency had a significant negative relationship with burden (*β* = −0.182, *p* < 0.05). In essence, as hope–agency increased, burden decreased. Model 3 revealed that hope–pathway did not have the same effect and was not significantly related to burden. See Table 4 for standardized OLS coefficients.

### 6.4. Hope Mediation Effects on Caregiver Distress and Burden Relationship

To address our third aim, mediation analyses were conducted. Because hope–pathway was not a significant predictor of burden when controlling for subjective stress, this variable was excluded from analysis and only hope–agency was tested.

Multiple linear regression models were used to estimate the paths of *a*, *b*, and $c^{\prime }$ as seen in Figure 1. The direct effect of subjective stress on burden is indicated by $c^{\prime }$. The indirect effect (mediation) of subjective stress on burden through hope–agency is indicated by *ab*. Results, which are reported in Figure 2, revealed that the direct effect of subjective stress on burden is $c^{\prime }$ = 1.008, while the indirect effect, mediated through hope–agency, is *ab* = (−0.118)(−0.796) = 0.094. The indirect effect accounts for 8.53% of the total effect and represents a partial mediation.

To further investigate the mediator, the bootstrapping method was utilized to examine whether hope–agency significantly mediated the effect of subjective stress on burden. The results showed that the 95% confidence interval of the indirect effect is (0.0094, 0.2324). Given that zero was not included in the interval, the indirect effect was statistically significant. Therefore, the mediation effect held.

## 7. Discussion

In the current study, we examined the relationships between stress, caregiver burden and two dimensions of hope; hope–agency and hope–pathway [23,34]. The findings of this study confirmed that the two dimensions of hope operate as independent resources in family caregivers of persons with Alzheimer’s disease. Among the individuals in our sample, hope–agency and hope–pathway was highly intercorrelated, but not colinear, suggesting that both hope dimensions may be present and should be attended to when working with caregivers. Moreover, both hope measures were independently and inversely correlated with burden among caregivers while hope–agency produced the higher-level inverse effect.

In this sample, caregivers who maintained a sense of confidence or self-efficacy in their ability to carry out their goal (hope–agency), as well as flexibility in problem solving and following the steps needed to achieve their goal (hope–pathway), experienced less burden than those who did not. Caregivers reporting higher levels of agency also indicated the capacity to find ways that supported their aspirations, possibly indicating the capacity of the agentic mindset to encourage wayfinding. The synergy between hope–agency and hope–pathway supports Snyder’s proposition around the agency and pathway interrelationship [23]. Our finding of a significant hope–agency and hope–pathway correlation provides some confirmation of this proposition. This pattern has also been found in non-caregiver samples [38]. Given these observations, interventions that promote agentic thinking and inform wayfinding may empower caregiver hopefulness and reduce the deleterious effects of burden.

Second, results from multivariate analyses of the relationship between burden and the two dimensions of hope revealed that hope–agency continued to significantly predict caregiver burden, even when controlling for other stressors, demographic variables, and social support. In contrast, even though hope–pathway was moderately correlated with caregiver burden, it did not significantly contribute to explaining burden once other key variables were controlled for. While hope–pathway or wayfinding was an important corelate of burden among the caregivers in this sample, its effect was largely eclipsed by other important stress process variables.

We surmise that the dimensions of hope may be understood as individual character strengths that were present before caregiving, and also as a dynamic state of mind that fluctuates depending on what a caregiver encounters each day. Therefore, people who are naturally more hopeful may have an affinity for facing some of the realities of providing care for a loved one with dementia. Given the long-term and unpredictable nature of caregiving, however, any caregiver can become less hopeful and experience burden.

It is also noteworthy that, in the current sample, caregivers who self-identified as female and viewed their situation as distressing or stressful were more likely to experience burden than caregivers who self-identified as male. Although this gender effect did not hold when hope–agency was taken into consideration, female caregivers may benefit from receiving additional support to build their sense of self-efficacy or agency in this role.

Finally, perhaps the most salient finding from this study is that hope–agency was a partial mediator between stress and burden in this sample of caregivers. Our findings confirmed a recent study documenting that a more general measure of hope was also a mediator between stress and burden in family caregivers [39]. These findings, together, build a case for testing clinical interventions aimed at addressing various aspects of hope in this population and examining the degree to which hope is modifiable and can lead to reducing or mitigating caregiver burden.

## 8. Implications and Recommendations

Although there are no specific interventions available to date which are focused specifically upon impacting caregiver hope–agency and hope–pathway in this popualtion, recommendations are provided from evidence-based interventions, which hold promise as worthy candidates for clinical trials that test their efficacy and utility in hope enrichment.

### 8.1. Hope-Focused Caregiver Interventions

A primary implication of this study for professionals who serve family caregivers is the relevance of using assessment tools and interventions for building and sustaining hope in this population with the goal of reducing perceived burden. While not specifically designed for caregivers of persons living with Alzheimer’s, there are hope-focused resources and interventions designed for other populations [40,41] that may have relevance for professionals and researchers interested in using hope-focused interventions for caregivers. The Oxford Handbook of Hope [41] provides a comprehensive review of the literature and research on conceptual frames and current hope-focused interventions. It critically examines prior applications in addition to discussing hope intervention studies. More recently, one systematic review of hope interventions identified thirty hope-related intervention studies involving palliative care patients [42], which may also be relevant for caregivers of persons with Alzheimer’s given that there is no cure. 

Weingarten [43] delivers a noteworthy hope-focused intervention for persons and families experiencing trauma, which we believe has applicability to caregivers of persons living with Alzheimer’s. This author offers the idea of reasonable hope, focusing professionals on goals and expectations that are, in the perception of the caregiver, attainable. Reasonable hope is … “consistent with the meaning of the modifier, suggests something both sensible and moderate, directing our attention to what is within our reach more than what may be desired but unattainable” [43]. This model offers a rare prescriptive for hope-enrichment work.

We purport that reasonable hope is more than a positive feeling. It is a present-centered practice focusing on dealing with what is currently available as a path for what the future brings. Rather than filling the time between the present and a possible outcome with anticipation, caregivers can attend to sense-making in the present rather than focus on an unknown future. Hope as a noun can be understood as a quantity, something that individuals possess to varying degrees, whereas hope as a verb is an ongoing interior process that a person can engage in. In this approach, professionals can accompany caregivers in their hope-challenged appraisals of the future and lostness in finding paths that lead to reasonable hope. Reasonable hope finding is a together act.

### 8.2. Caregiver Interventions That Indirectly Promote Hope

Recent systematic reviews outline an array of multi-component psychoeducation and psychotherapeutic interventions focused on reducing burden and improving caregiver well-being [44] including internet based options [45]. A few examples are Coping with Caregiving for culturally diverse caregiving populations [46]; the Minnesota adaptation of the NYU caregiver intervention for adult children caregivers [47]; and REACH II [48]. There are also telehealth, video, and internet asynchronous and synchronous delivery platforms such as Tele-Savvy Caregiver [49]. We hypothesize that these multicomponent interventions may indirectly promote hope among caregivers. Additionally, we hypothesize that an addition of a hope-focused supplemental component to these interventions may serve specific caregivers who are experiencing a high degree of hopelessness.

Case management and advocacy are macro-level interventions that address access to care, structural racism, and economic justice issues that may increase burden and impact caregiver hope. Effective dementia care management support may provide caregivers with alternative pathways in dealing with the complexities of access to, and engagement with, the community health and human services continuum. Having access to this level of professional expertise and advocacy may empower caregivers with tangible support that energizes hope–agency and removes barriers to hope-driven wayfinding.

### 8.3. Future Research on Caregiver Hope and Burden Is Needed

The current study provides evidence that caregiver burden can be mediated by hope-agency. However, the empirical analyses conducted in this study do not clarify how caregivers formulate agency (their vision of a desirable future) or how they discern pathways toward this future, given the vicissitudes and uncertainties associated with caregiving. Additionally, this study did not examine the role of caregiver self-efficacy as it relates to hope and burden, which is an important and related arena for future research. In-depth, qualitative and quantitative studies are needed to specify how hope–agency and hope–pathway find expression as they enact the caregiver role over time (i.e., at different points in disease progression) particularly given that Alzheimer’s disease is a progressive disease without a cure. Likewise, this research will be important for understanding more about how hope is conceptualized within palliative care contexts related to progressive neurocognitive diseases such as Alzheimer’s disease.

Refinements in research design, data gathering strategies, and sampling considerations will improve understanding of the caregiver hope and burden relationship. Clarification of feedback loops among burden correlates, caregiver hope dimensions, and caregiver burden will enlighten understanding of the interactive processes at play as caregivers negotiate caregiving in the context of the ever-changing context of disease progression in Alzheimer’s disease and other forms of progressive dementia.

## 9. Limitations

This convenience sample was limited to caregivers living within a private home in community settings, usually residing with their loved one with Alzheimer’s in a large metropolitan area of the U.S. Effective future research and interventions aimed at activating the benefits of hopefulness for mitigating caregiver burden will require a more finessed understanding of a caregiver’s cultural, ethnic, and/or religious/spiritual beliefs and practices. More specifically, studies need to be conducted in rural and frontier regions of the world as there is a lack of knowledge of how hope operates in caregivers who reside in less populated areas, particularly Indigenous or First Nation peoples.

Additionally, a cross-sectional design was used in this study which limited what we could learn about caregivers’ experiences over time. Longitudinal studies will serve to validate the extent to which “preexisting” dispositional caregiver hope directly impacts burden outcomes and the directionality of these relationships. A longitudinal approach will also allow for an examination of caregiver hope in different stages of disease progression and how hope is manifest on a day-to-day basis in caregivers.

In addition to adopting a trait–hope assumption, this research did not account for the notion of hope variability within life domains (relational, academic, work, and leisure), as proposed by Sympson [50]. Rather than assume that caregivers’ level of hope–agency and hope–pathway was based on the challenges of caregiving, it is also important to better understand the intersectionality of the caregiving role with other life domains.

## 10. Conclusions

There are noteworthy contributions from this study of the relationship between stress, hope, and burden among family caregivers of persons with Alzheimer’s. Caregivers may be engaged in a continuous future-oriented, cognitive appraisal process, involving a reciprocal and additive exchange between hope–agency ends and hope–pathway means. There is evidence that caregiver hope is individual and dispositional but may also be modifiable based on their adaptation and learning new responses to continuous feedback along the temporal pathway experience.

Hope–agency, in this study, was a mediator of the relationship between caregiver stress and the extent of perceived burden. These findings affirm a recent study [27] on the explanatory power of hope for predicting caregiver burden while being among the first to conceptually center our measurement approach within the widely applied and validated Snyder [34] concept of hope measure dimensionality.

We offer applied implications of our findings, intended to inform researchers and professionals seeking to reduce caregiver burden by testing interventions intended to realistically strengthen the caregiver sustainment of confidence in future possibilities (hope–agency) and to identify viable processes and pathways for movement toward these possibilities (hope–pathway).

## Figures and Tables

**Figure 1 geriatrics-09-00038-f001:**
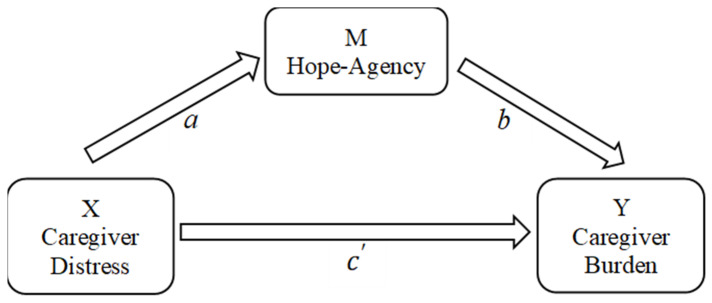
Proposed mediation model.

**Figure 2 geriatrics-09-00038-f002:**
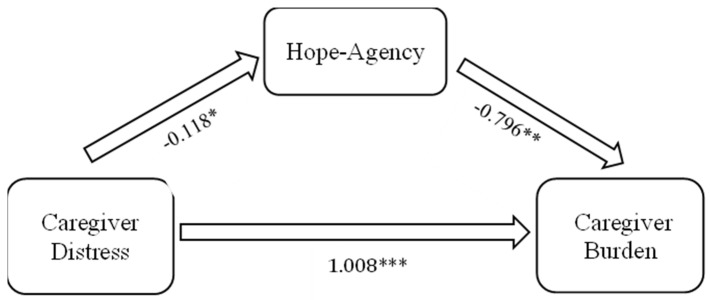
Results of mediation model. * *p* < 0.05. ** *p* < 0.01. *** *p* < 0.001.

**Table 1 geriatrics-09-00038-t001:** Demographic information for sample.

	Mean (SD)	*n* (%)
Caregiver age in years (*n* = 149)	64.8 (11.1)	
Caregiver self-reported gender (*n* = 153)		
Male		45 (29.4)
Female		108 (70.6)
Caregiver education (*n* = 151)		
High School or equivalency		11 (7.2)
Undergraduate		97 (64.3)
Graduate		40 (26.5)
Other		3 (2.0)
Caregiver ethnicity (*n* = 152)		
Hispanic or Latino		5 (3.3)
Not Hispanic or Latino		147 (96.7)
Caregiver race (*n* = 153)		
Person of Color		8 (5.2)
White or Caucasian		145 (94.8)
Relationship to person with dementia (*n* = 151)		
Spouse		100 (66.2)
Adult child/grandchild		35 (23.2)
Other		16 (10.7)

**Table 2 geriatrics-09-00038-t002:** Descriptive statistics for measures.

	N	Range	Mean	SD	Alpha
Outcome					
Caregiver Burden	138	3–83	30.9	15.3	0.93
Primary Stressors					
Objective (Intensity of Care)	146	0–8	3.3	2.3	N/A *
Subjective (Caregiver Distress)	116	0–38	8.9	8.7	0.85
Coping Resources					
Social Support	151	6–20	17.1	2.7	0.78
Hope-Agency	117	5–28	21.4	4.4	0.82
Hope-Pathway	117	7–28	21.3	4.1	0.84

Note: * Intensity of Care is an additive index representing the sum of hours spent providing care for the person with dementia and the number of total people cared for. Therefore, an alpha was not calculated for this index.

**Table 3 geriatrics-09-00038-t003:** Correlation matrix for variables.

Variable	1	2	3	4	5	6	7	8	9
1. Burden	_______								
2. Objective Stress	0.23 **	_______							
3. Subjective Stress	0.62 **	0.04	_______						
4. Age	−0.10	0.14	−0.21 *	_______					
5. Gender	0.27 **	0.02	0.10	−0.20 *	_______				
6. Marital Status	−0.07	0.07	−0.10	0.12	−0.06	_______			
7. Social Support	−0.22 *	0.03	−0.17	0.02	0.08	−0.06	_______		
8. Hope-agency	−0.33 ***	0.07	−0.26 **	0.11	−0.06	−0.13	0.44 ***	_______	
9. Hope-pathway	−0.24 **	0.12	−0.23 **	0.08	−0.00	0.01	0.32 ***	0.61 ***	_______

Note: * *p* < 0.05. ** *p* < 0.01. *** *p* < 0.001.

**Table 4 geriatrics-09-00038-t004:** Standardized Ordinary Least Squares (OLS) coefficients predicting caregiver burden.

	Model 1	Model 2	Model 3
Primary Stressors			
Objective (Intensity of Care)	0.142	0.157 *	0.158 *
Subjective (Caregiver Distress)	0.566 ***	0.527 ***	0.542 ***
Secondary Stressors			
Caregiver Age	0.027	0.039	0.030
Caregiver Gender	0.266 ***	0.249 ***	0.262 ***
Marital Status	−0.021	−0.049	−0.023
Resources			
Social Support	−0.143	−0.065	−0.108
Hope-Agency		−0.182 *	
Hope-Pathway			−0.097
Intercept	18.740	26.967	22.948
*F*	15.036	14.033	13.133
*R* ^2^	0.49	0.51	0.49
∆*R*^2^		0.02 *	0.00
N	102	102	102

Note: * *p* < 0.05. *** *p* < 0.001.

## Data Availability

The data presented in this study are not publicly available for the purposes of privacy and confidentiality of research participants.

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
