# Peer review of "Hope Mediates Stress to Reduce Burden in Family Caregivers of Persons with Alzheimer’s Disease"

_geriatrics, 2024, doi:10.3390/geriatrics9020038_

Round 1

Reviewer 1 Report

Comments and Suggestions for Authors

The aim of the paper is to present research findings regarding the role of hope-agency and hope-pathway in caregiver burden from a study of caregivers utilizing multiple dimensions of measurement. The paper offers valuable contributions to those who serve caregivers of persons with dementia and a further understanding of the phenomenon of hope and how it can positively support caregivers and persons with dementia’s quality of life, ability to age in place, and overall well-being. The correlation of hope-agency to self-efficacy and hope-pathway to activation builds upon and applies to health sciences research regarding engagement in personal health and positive outcomes. The paper is well-written, easy to follow, and thoroughly examines hope and the hope-agency model.

  • General concept comments
    Article: A few areas of missing information were identified, and these are noted in the specific comments section.
    Review: The manuscript offers new knowledge on the connection of hope-agency and hope-pathway with caregiver burden, which is valuable in designing interventions and services for caregivers and supporting persons with dementia to live in the lowest level of care.
  • Specific comments
  • Abstract Line 20: Maybe rephrase to state “multiple dimensions of hope” rather than multiple hope dimensions.
  • Line 36: You use the reference Thompson et al. 2005. Is there something more recent about caregiving's physical, emotional, social, and spiritual health costs? Or is this a seminal piece of work? You use this again on line 47.
  • Line 38:  You state, “Prior research on hope and caregiving has considered hope from a unidimensional perspective.” This is a strong statement. Has all prior research done this? Or should it say that most prior research has focused on a unidimensional perspective?
  • Line 56: Rephrase to state that it has been suggested that increased mortality rates may be related to an increase in harmful behaviors on the caregiver's part for clarity.
  • Line 61: Rephrase to include a description of social services, such as social service support, with a couple of examples or something like that for clarity. As it is the reader may not know how social services allow for a person with dementia to live in the homes of their loved ones. For example, are you talking about personal care services, social work services, home health services, etc.
  • Line 130: Does “in the caregiver” need to be added to the end of this sentence? As it is, I am unsure if you are talking about serious untreated mental illness in the caregiver or the person with dementia.
  • Section 5.2.2 Predictor variables: Please include consistency and reliability for all variables. In the outcome variable section above, you have both consistency and reliability for the outcome variable, and then for the predictor variables, you sometimes include consistency and then sometimes include reliability. If only one is included, please explain why you are not including the other measure. The intensity of care variable does not include either. I am assuming you ran initial consistency and reliability on this pilot measure. Since the secondary stressor variable really is demographics only, these measures would not be needed on that variable.
  • Table 2: Why is the Alpha N/A for the Objective stressor measure? This should be noted somewhere.
  • Line 305: You indicate that the hope pathway is modifiable based on study findings. I am not clear on this connection to the study. Please expand.
  • Line 386-387: You mention the study was in a large metropolitan area, which is a limitation. I suggest that mentioning that future research is conducted in rural and frontier areas is appropriate.

Author Response

Please see attachment. Thank you very much for your careful and insightful review.

Reviewer 2 Report

Comments and Suggestions for Authors

This article reports on an adequately powered study on associations of multiple dimensions of hope in caregivers of persons with dementia. Indeed, positive caregiving experiences has been the topic of research more recently, adding a relevant perspective to the large body of literature on family caregiver burden. This is a fascinating, relevant topic. Use of theory, distinguishing hope agency and hope pathway, hypothesizing mediation speaks to the soundness of the research design. There is an elaborate discussion of literature and available interventions. Below are some points to improve the paper.

Abstract: you may wish to include some figures, such as on strength of the associations which will add to credibility of your results when reading the abstract.

Introduction: the first sentence (There are approximately 6.7 million people over the age of 65 in the United States….) does not appeal to an international audience and is boring as it is well know that the population of persons with dementia is large and growing.  Next para starts with costs and only then you get to the topic of hope.

Methods: as the sources are blinded, I cannot evaluate whether the protocol of this study was pre-registered, and to what extent the reported analyses had been planned.

Methods: Caregiver stress was measured by the NPI which refers to stress caused by challenging behaviour only. Multiple other sources of stress exist. Intensity of care was measured as an objective measure of stress, but intensity may not equate to caregivers stress in a linear fashion. Isn’t caregiver stress a subjective experience, and intensity of care another concept. The results, the correlation is modest at best. Better just call it intensity of care rather than a measure of burden.

Line 186 no need to refer to the Results tables in the Methods section

Table 1: limit to 1 decimal (age: 64.76 (11.06)) also in Table 1, 2 decimals suggest an accuracy that is not appropriate

Table 3 is confusing on first sight with columns indicated with numbers up to 8, and rows with variables numbered up to 9.

Table 4. explain abbreviation OLS in a footnote to the table

Line 253. Please move from Results to Methods: “Since the Sobel test is conservative (MacKinnon et al., 1995), an increasingly popular method of testing the indirect effect is bootstrapping (Shrout & Bolger, 2002).

Line 259: Please include moderation analyses in the Methods, including your hypotheses, and in the Results indicated which moderators were examined (the two hope items?) “There was no evidence of a moderation effect.”

You may reflect upon the cross-sectional design of the study; currently it is hidden in a recommendation for future research to conduct longitudinal studies.

You may wish to reflect on possible overlap between how hope is being measured, and self-efficacy. Further, do the hope measures capture all dimensions of hope of those confronted with a progressive disease without cure? With hope in family caregivers? With hope as conceptualized in palliative care?

Author Response

Please see attachment. Thank you for your careful and generous review of this manuscript.
